



# Evaluation of interactive and prescribed agricultural ammonia emissions for simulating atmospheric composition in CAM-Chem

Julius Vira[1,2], Peter Hess[1], Money Ossohou[3,4], and Corinne Galy-Lacaux[5]

[1]Department of Biological and Environmental Engineering, Cornell University, Ithaca, NY, USA
[2]Finnish Meteorological Institute, Helsinki, Finland
[3]University of Man, Man, Côte d'Ivoire
[4]Laboratoire des Sciences de Matière, de l'Environnement et de l'Energie Solaire, Université Félix Houphouët-Boigny, Abidjan, Côte d'ivoire
[5]Laboratoire d'Aérologie, Université de Toulouse, CNRS, Observatoire Midi Pyrénées, Toulouse, France

**Correspondence:** Julius Vira (julius.vira@gmail.com)

**Abstract.** Ammonia ($NH_3$) plays a central role in the chemistry of inorganic secondary aerosols in the atmosphere. The largest emission sector for $NH_3$ is agriculture, where $NH_3$ is volatilized from livestock wastes and fertilized soils. Although the $NH_3$ volatilization from soils is driven by the soil temperature and moisture, many atmospheric chemistry models prescribe the emission using yearly emission inventories and climatological seasonal variations. Here we evaluate an alternative approach

where the $NH_3$ emissions from agriculture are simulated interactively using the process model FANv2 (Flow of Agricultural Nitrogen, version 2) coupled to the Community Atmospheric Model with Chemistry (CAM-chem). We run a set of six-year global simulations using the $NH_3$ emission from FANv2 and three global emission inventories (EDGAR, CEDS and HTAP) and evaluate the model performance using a global set of multi-component (atmospheric $NH_3$ and $NH_4^+$, and $NH_4^+$ wet deposition) in-situ observations. Over East Asia, Europe, and North America, the simulations with different emissions perform

similarly when compared with the observed geographical patterns. The seasonal distributions of $NH_3$ emissions differ between the inventories, and the comparison to observations suggests that both FANv2 and the inventories would benefit from more realistic timing of fertilizer applications. The largest differences between the simulations occur over data-scarce regions. In Africa, the emissions simulated by FANv2 are 200–300 % higher than in the inventories, and the available in-situ observations from Western and Central Africa, as well as $NH_3$ retrievals from the IASI instrument, are consistent with the higher $NH_3$

emissions as simulated by FANv2. Overall, in simulating ammonia and ammonium concentrations over regions with detailed regional emission inventories, the inventories based on these details (HTAP, CEDS) capture the atmospheric concentrations and their seasonal variability the best. However these inventories can not capture the impact of meteorological variability on the emissions, nor can these inventories couple the emissions to the biogeochemical cycles and their changes with climate drivers. Finally, we show with sensitivity experiments that the simulated time-averaged nitrate concentration in air is sensitive

to the temporal resolution of the $NH_3$ emissions. Over the CASTNET monitoring network covering the U.S., resolving the $NH_3$ emissions hourly instead monthly reduced the positive model bias from approximately 80 % to 60 % of the observed yearly mean nitrate concentration. This suggests that some of the commonly reported overestimation of aerosol nitrate over the U.S. may be related to unresolved temporal variability in the $NH_3$ emissions.



## 1 Introduction

Volatilization of ammonia ($NH_3$) from fertilizers and livestock wastes constitutes the largest source of atmospheric $NH_3$. Once emitted to the atmosphere, $NH_3$ reacts with oxides of sulfur and nitrogen to form secondary aerosols. Ammonia is often the limiting precursor for formation of ammonium nitrate, which is projected to become an increasingly important aerosol

component with impacts on air quality and aerosol radiative forcing (Bauer et al., 2007; Hauglustaine et al., 2014). Similar to other reactive nitrogen species, ammonia may have adverse ecological effects when deposited on sensitive ecosystems (Duprè et al., 2010; Payne et al., 2017).

In Vira et al. (2020), we introduced an updated version (FANv2) of the process model FAN (Flows of Agricultural Nitrogen), which simulates the physical mechanisms of ammonia volatilization interactively within an Earth system model. Incorporated

into the Community Land Model (CLM), the land component of the Community Earth System Model (CESM), FANv2 evaluates $NH_3$ emissions arising from fertilizer use, grazing livestock, and manure management. Here we couple FANv2 and CLM with the Community Atmosphere Model (CAM-chem;  Lamarque et al., 2012; Tilmes et al., 2015) to evaluate how interactive $NH_3$ emissions affect the simulation of atmospheric composition and nitrogen deposition. Distinct from the earlier efforts (Bash et al., 2013; Zhu et al., 2015; Shen et al., 2020) to interactively simulate the $NH_3$ emissions in atmospheric chemistry

models, we include the $NH_3$ emissions from both fertilizer applications and livestock manure. Including manure as an $NH_3$ emission source increases the fraction of emissions resolved by the process model considerably, since globally about 60–80 % of the total agricultural $NH_3$ emissions are estimated to originate from manure (Beusen et al., 2008; Paulot et al., 2014; Vira et al., 2020).

The emission of $NH_3$ is sensitive to soil temperature and moisture (Fenn and Hossner, 1985; Sommer et al., 2004), and

evaluating the $NH_3$ volatilization interactively through a process model can help resolve both long and short-term emission variations driven by the meteorological forcing and agricultural activities, which are often linked to meteorology (e.g., planting dates). The emission inventories, on the other hand, are able to more easily incorporate details regarding regional fertilization and manure management practices, which also affect timing and magnitude of the emissions. However, the representation of these practices in the inventories is largely static. The present paper therefore aims to evaluate the tradeoff between the

representation of process-level details in FANv2 and the representation of agricultural practices in the emission inventories. We evaluate the performance of FANv2 against measurements in simulating atmospheric gas-phase ammonia, ammonium ($NH_4^+$) and nitrate ($NO_3^-$) aerosols and the wet deposition of ammonium. The CAM-chem simulations using ammonia emissions from FANv2 are compared to simulations based on three global, state-of-art ammonia emission inventories and evaluated using data from atmospheric observing networks covering parts of Africa, East Asia, Europe and the United States. The comparison for

stations located in Africa is especially interesting, since as shown in Vira et al. (2020), the $NH_3$ emissions for Africa simulated by FANv2 are several times higher than previous estimates.

The volatility of $NH_3$ in soils and the thermodynamic stability of ammonium nitrate in aerosols are oppositely affected by air temperature. While the temperature dependence of ammonium nitrate aerosols is usually included in atmospheric models, the temperature dependence of soil ammonia emissions is usually not represented, and the anticorrelation between the $NH_3$





emissions and the nitrate formation is therefore not reproduced. We hypothesize that resolving the $NH_3$ emissions' response to the meteorological forcing might reduce the positive bias in simulated airborne nitrate concentrations, which has been reported for some regions in both CAM (Lamarque et al., 2012) and other models (Heald et al., 2012; Walker et al., 2012; Paulot et al., 2016). We test this hypothesis by running a set of model experiments.

## 2   Methods

The model runs are performed using the Community Earth System Model (CESM) version 2. Only the land and atmospheric components (CLM and CAM-chem) are active, while the ocean and sea ice are prescribed. The $NH_3$ FANv2 emissions in the CLM were analyzed previously in Vira et al. (2020), and here we focus on how the emissions affect the simulated atmospheric composition. Specifically, we compare the atmospheric concentrations in CAM-chem using the $NH_3$ emissions generated in
FANv2 with three additional prescribed emission inventories.

### 2.1   The FANv2 process model

The FANv2 process model and its input data are described in Vira et al. (2020). The model evaluates $NH_3$ emissions from both synthetic fertilizers and livestock manure. Ammonia volatilization from livestock manure is evaluated separately for emissions from pastures, animal housings and storage and the spreading of the manure. The emissions from synthetic fertilizers are
evaluated separately for urea and other fertilizers.

The $NH_3$ flux from soils is evaluated using a single-layer resistance scheme which includes the partitioning between gaseous, aqueous and adsorbed phases of ammonia and ammonium and evaluates the solute and gas diffusion in the water and air filled soil pores. The emissions from animal housings and manure stores are evaluated using the parameterization of Gyldenkærne et al. (2005).

The global manure nitrogen excretion (120 Tg N $yr^{-1}$) was derived from livestock data sets (Robinson et al., 2011, 2014) released by the United Nations Food and Agriculture Organization (FAO) using the nitrogen excretion rates given in IPCC (2006). The synthetic fertilizer use (79–87 Tg N $yr^{-1}$ for 2010–2015) was based on the Landuse Harmonization 2 dataset (Lawrence et al., 2016; Hurtt et al., 2020) prepared for use within the the Coupled Model Intercomparison Project Phase 6 (CMIP6).

### 2.2   The Community Land Model

The FANv2 process model was introduced into the CLM version 5 (Lawrence et al., 2018), which forms the terrestrial component of the CESM version 2. The CLM evaluates the soil conditions (temperature, moisture, rainwater infiltration) and the aerodynamic and quasi-laminar layer resistances required by FANv2. CLM also includes a representation of the terrestrial nitrogen (N) cycle and its effects on carbon cycling and vegetation. In this study, the simulated N losses in FANv2 are not
propagated back to the N cycling in the CLM, and the amounts of fertilizer N available to crops are therefore not affected by FANv2.




The timing of synthetic fertilizer application is determined by the CLM crop model (Levis et al., 2012, 2018; Lombardozzi et al., 2020), which includes phenological parameterizations for 8 major crops, which may coexist in the same grid cell within the CLM sub-grid structure. Synthetic fertilizer is applied in FANv2 over a single 20-day window following the simulated time of leaf emergence, which is determined mainly by thresholds dependent on growing degree days and air temperature. The

timing of the synthetic fertilization, and consequently the fertilizer $NH_3$ emissions, therefore depends on both crop types and meteorological conditions.

## 2.3   The Community Atmospheric Model with Chemistry

The emission flux of ammonia, as evaluated by the FANv2 model within the CLM, is passed to the Community Atmospheric Model with interactive chemistry (CAM-chem). The CAM-chem version 5.4 was run in the "offline" configuration (Lamarque

et al., 2012), where the meteorological fields were prescribed from the MERRA reanalyses (Rienecker et al., 2011) from 2010–2015. The formation of ammonium sulfate and nitrate aerosols is simulated by the Bulk Aerosols Model (BAM) following the scheme of Metzger et al. (2002), which parameterizes the thermodynamic equilibrium between sulfuric and nitric acid, ammonia, and ammonium in gaseous and aerosol phases.

Ammonia and ammonium are removed from the atmosphere through wet and dry deposition. Dry deposition of ammonia is

evaluated using the resistance approach of Wesely (1989) with updates described by Emmons et al. (2010); dry deposition of aerosols is described in Lamarque et al. (2012). The wet deposition of soluble gases including ammonia is simulated with the algorithm of Neu and Prather (2012), while aerosol wet deposition is parameterized according to Barth et al. (2000).

## 2.4   Simulations

Four CESM simulations for 2010–2015 were run. In the first simulation the CESM is run with the FANv2 $NH_3$ emissions

coupled to CAM-chem. An additional three simulations are run with agricultural $NH_3$ emissions taken from three global emission inventories for 2010: the Emissions Database for Global Atmospheric Research version 4.3.2 (EDGAR;  Crippa et al., 2018), the HTAP_v2.2 inventory (Janssens-Maenhout et al., 2015), and the Community Emissions Data System (CEDS; Hoesly et al., 2018). Both the CEDS and HTAP_v2.2 inventories are based on merging the global emissions from EDGAR with more detailed regional inventories where available. In CEDS, the regional inventories are introduced by scaling the EDGAR

emissions to a country level, while in HTAPv2.2, the regional inventories are merged into EDGAR in gridded form. The total $NH_3$ emissions in the CEDS and HTAP_v2.2 inventories are rather similar; both CEDS and HTAP_v2.2 emit 35–36 Tg $NH_3$-N yr$^{-1}$ from agriculture, which is 6–7 Tg N less than EDGAR (42 Tg N), and 12–13 Tg N less than FANv2 (48 Tg N, Table 1). Regionally, the largest differences between FANv2 and the inventories occur over Africa, where FANv2 emissions are about 5 Tg N higher, and over Latin America, where the FANv2 emissions are 3–3.5 Tg N higher. The $NH_3$ emissions in each inventory

are given on a monthly time resolution, while the FANv2 emissions are evaluated on each model timestep.

The simulations differ only in the $NH_3$ emissions from the agricultural sector; all other anthropogenic emissions, including 5.4 Tg $NH_3$-N from non-agricultural sources, are taken from the HTAPv2.2 inventory. Biogenic and biomass burning emissions





**Table 1.** Agricultural ammonia emissions by region in the FANv2 simulation and in the CEDS, EDGAR 4.3.2 and HTAPv2.2 inventories. The FANv2 emissions are averaged for 2010–2015; the other emissions are for the year 2010. The totals are interpolated from gridded emissions; the European emissions include the part of Russia west of $60°$ E.

| Region | $NH_3$ emission, Tg N yr$^{-1}$ | | | |
|---|---|---|---|---|
| | CEDS | EDGAR | HTAP | FANv2 |
| Africa | 2.1 | 2.4 | 2.1 | 7.2 |
| Asia except China and India | 5.4 | 7.5 | 5.4 | 8.2 |
| China | 7.8 | 11.6 | 6.9 | 7.5 |
| Europe | 3.8 | 5.4 | 3.5 | 4.8 |
| India | 6.0 | 5.5 | 6.2 | 7.5 |
| Latin America | 4.3 | 4.7 | 4.0 | 7.4 |
| North America | 3.2 | 3.6 | 3.0 | 3.5 |
| Oceania | 0.6 | 0.6 | 0.5 | 1.4 |
| World | 36.2 | 41.6 | 34.7 | 47.6 |

are taken from the CAM-chem default input dataset (Lamarque et al., 2012); this includes $NH_3$ emitted from natural soils (2 Tg N yr$^{-1}$), oceans (6.7 Tg N yr$^{-1}$) and biomass burning (4.7 Tg N yr$^{-1}$).

Four additional one-year (2010) simulations were run to investigate the effect of the temporal resolution of the $NH_3$ emissions on nitrate aerosols. The $NH_3$ emissions from FANv2 were used in these simulations, but instead of running FANv2 interactively, the emission from the FANv2-based simulation for 2010 was averaged to hourly, daily, monthly and finally yearly time resolutions and then used as input to CAM-chem.

Both CAM and CLM were run in a global longitude-latitude grid with a 2.5×1.9 degree resolution and a 30 minute coupling time step. The year 2009 was run for spin-up.

## 2.5 Observations

The simulations are evaluated against data from various atmospheric monitoring networks. While the $NH_3$ is emitted in gaseous form, gas-phase $NH_3$ has a short atmospheric lifetime making the observations sensitive to local sources or sinks. To obtain a more robust picture of how the different emission inventories perform in the CAM-chem simulations, we also include observations of particulate $NH_4^+$ and $NH_4^+$ wet deposition. In addition, observations of particulate nitrate ($NO_3^-$) are used to evaluate the effect of $NH_3$ emissions on nitrate aerosols.

We use data from 6 networks (Table 2): the National Trends (NTN, http://nadp.slh.wisc.edu/ntn/), Clean Air Status (CAST-NET, https://www.epa.gov/castnet) and Ammonia Monitoring (AMoN, http://nadp.slh.wisc.edu/amon/) networks for North America, the European Monitoring and Evaluation Programme (EMEP, https://www.emep.int/) network for Europe, the International Network to study Deposition and Atmospheric composition in Africa (INDAAF, https://indaaf.obs-mip.fr/) for Africa, and the Acid Deposition Monitoring Network in East Asia (EANET, https://www.eanet.asia/) network for Eastern Asia. All



**Table 2.** Summary of the atmospheric monitoring network datasets used in this study. The number of stations shown is evaluated after screening for completeness (see text for details).

| Region | Network | Species | Original resolution | # stations |
|---|---|---|---|---|
| Africa | INDAAF | $NH_3$ (air) | Monthly | 7 |
| | | $NH_4^+$ (air) | Weekly[1] | 4 |
| | | $NO_3^-$ (air) | Weekly[1] | 4 |
| | | $NH_4^+$ (rain) | Daily[2] | 5 |
| Europe | EMEP | $NH_3$ (air) | Various[3] | 47 |
| | | $NH_4^+$ (air) | | 51 |
| | | $NO_3^-$ (air) | | 64 |
| | | $NH_4^+$ (rain) | | 57 |
| East Asia | EANET | $NH_3$ (air) | Monthly[4] | 16 |
| | | $NH_4^+$ (air) | | 15 |
| | | $NO_3^-$ (air) | | 16 |
| | | $NH_4^+$ (rain) | | 28 |
| U.S. | AMoN | $NH_3$ (air) | Bi-weekly | 33 |
| | CASTNET | $NH_4^+$ (air) | Weekly | 81 |
| | CASTNET | $NO_3^-$ (air) | Weekly | 81 |
| | NTN | $NH_4^+$ (rain) | Weekly | 219 |

[1]One 24 h exposure per week for 2000–2004, 7-day exposures since 2005; [2]Reported per precipitation event; [3]Hourly to weekly; [4]Data obtained from monthly summaries.

model-measurement comparisons are performed either on a monthly or multi-annual basis. The comparisons of wet deposition are based on measured ammonium concentrations in rainwater, which are converted to monthly wet deposition by multiplying by the observed precipitation reported with the data. The modeled wet deposition is evaluated as the sum of the scavenged $NH_4^+$ and $NH_3$.

5     All observations were first screened using the quality flags provided with the data, averaged to the monthly level and then checked for temporal coverage. Due to the seasonal variation of the $NH_3$ emissions, the criteria for temporal coverage prioritize seasonal completeness, whereas the required number of years within the simulated period was adjusted for each network in order to retain sufficient spatial representativeness. This was implemented as follows:

1. Months with less than 50 % of days covered were omitted

10    2. Years with less than 10 valid months were omitted

3. For the INDAAF network, stations with less than one valid year (according to steps 1 and 2) after the year 2000 were omitted





4. For the EMEP network, stations with less than 2 valid years within the 2010–2015 period were omitted

5. For the remaining networks, stations with less than 3 valid years within 2010–2015 were omitted.

The monthly coverage test (item 1 above) was generally evaluated using the averaging period of each observation. However, this criterion could not be applied to the automatic precipitation samplers in the INDAAF network, which sample discrete precipitation events. In earlier studies (Yoboué et al., 2005; Galy-Lacaux et al., 2009) the average collection efficiency of these samplers was 85–90 % of the total precipitation. Furthermore, the INDAAF aerosol observations ($NH_4^+$ and $NO_3^-$) collected before 2005 consists of one-day filter exposures repeated every 7 days. These measurements were taken over each 7-day window so that 2–3 samples were required for each valid month. Finally, the INDAAF dataset includes little data after 2010, and therefore all data from 2000 and later years were included. The simulations were compared with the observations only as temporal averages over the simulated and measured periods.

For the other networks, the observations were extracted for 2010–2015 and compared both annually averaged and on a monthly basis. The EANET dataset includes urban stations, which were omitted due to questions concerning their representativeness at the 2 degree model resolution; the other networks include mainly rural or remote sites. The EMEP network includes wet deposition observations collected using bulk samplers, which in some conditions overestimate the wet deposition flux due to contamination from dry deposition (Vet et al., 2014). In this paper, the bulk sampler measurements are presented visually but not included in the computation of the measurement statistics.

Fewer data are available to constrain the ammonia and ammonium outside the regions covered by the networks listed above. In Section 3.1.2 we compare the simulated wet deposition with observed averages compiled from published literature (Table A1). These observations cover one or more years between 2000 and 2013, depending on the study; all comparisons are made with respect to the simulated 2010–2015 average.

Finally, we compare the simulated column-integrated $NH_3$ concentrations with the dataset of by Van Damme et al. (2018b), which consists of $NH_3$ retrievals from the IASI (Infrared Atmospheric Sounding Interferometer) instrument onboard the MetOp satellites. The IASI data shown here are obtained from the oversampled level 3 dataset (Van Damme et al., 2018a), which consists of $NH_3$ column densities retrieved for the morning overpasses averaged over the years 2008–2016; only the observations of $NH_3$ total columns that have 10 % or less cloud cover are used. A detailed description of the algorithm can be found in Van Damme et al. (2017). A quantitative comparison between IASI and the model results would require a careful consideration of temporal sampling and vertical sensitivity of the retrievals, and our comparisons are therefore mainly qualitative and complement the in-situ observations over data-sparse regions.

## 3  Results and discussion

The global distribution of ammonia and ammonium (Fig. 1) reflect the global emission patterns and highlight India, Northern China and the Western Europe as the regions most impacted by ammonia emissions. Comparison with observations shows that the spatial patterns of particulate ammonium and the $NH_4^+$ wet deposition are generally well captured in FAN and the

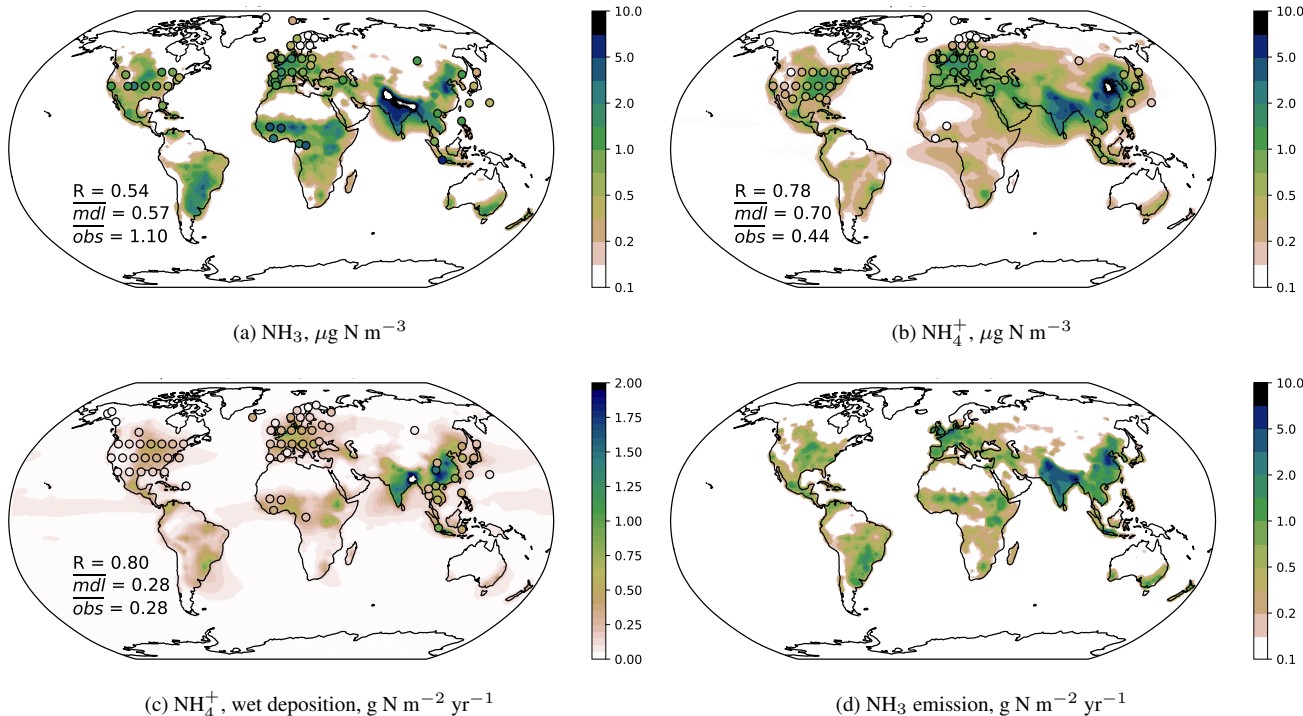

**Figure 1.** Simulated global distribution for 2010–2015 of near-surface level ammonia and ammonium ($\mu$gN m$^{-3}$, panels a and b), wet deposition of ammonium and emission of ammonia (gN m$^{-2}$ yr$^{-1}$ panels c and d) in FANv2. Markers indicate observed values. For the sake of clarity, the density of observations has been reduced by averaging into a 8-degree grid where necessary. The correlation coefficient ($R$), average of observations, and model average at observed sites are shown for panels a-c. The statistical parameters are evaluated after spatial averaging to the 8-degree grid. The NH$_4^+$ wet deposition includes the wet deposition of gas-phase ammonia. The maps for the CEDS, EDGAR and HTAP simulations are shown in Suppl. Figs. S1–S3.

other simulations ($R$ =0.7–0.8; Figs. S1–S3) on the global scale. The observed pattern of NH$_3$ is reproduced less accurately ($R$ =0.3–0.5). The smaller correlation for NH$_3$ is in part explained by the shorter atmospheric lifetime of NH$_3$, which results in spatial gradients that cannot be reproduced at the 2-degree resolution. However, the comparisons also indicate that the model tends to simultaneously underestimate NH$_3$ and overestimate NH$_4^+$ regardless of the NH$_3$ emission inventory used.

5    Fig. 2 compares the FAN and EDGAR simulations to the retrievals of column-integrated NH$_3$ from the IASI instrument. Consistent with the comparison with surface observations of NH$_3$, both the FAN and EDGAR simulations are biased low in parts of the East Asia, Eastern Europe and the Central U.S. The largest differences occur over Africa, India and South America, which will be discussed in Sections 3.1.1 and 3.1.2.

In the following section, we analyze the simulated atmospheric NH$_3$ and NH$_4^+$ and their wet deposition regionally. The
10   simulation of nitrate aerosols is discussed separately in Section 3.2. The model evaluations are summarized by Taylor plots,



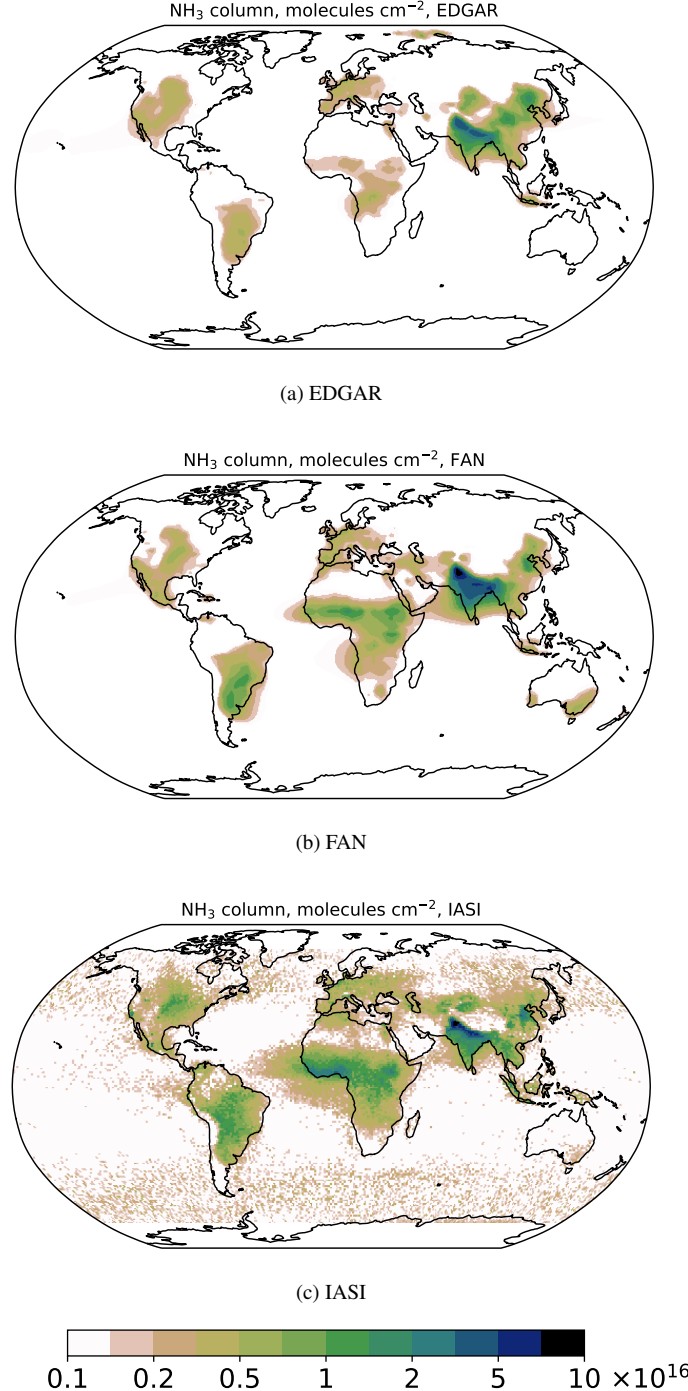

**Figure 2.** Column density of $NH_3$ (molecules $cm^{-2}$) in the (a) EDGAR and (b) FAN simulations for 2010–2015 and (c) in the IASI dataset (Van Damme et al., 2018b) averaged over 2008–2016. The results for CEDS and HTAP are shown in Fig. S5.



which were calculated separately in space (averaged in time, comparing spatially across stations) and time (averaged each month over all stations, comparing the timeseries). The comparisons are also presented as scatter plots in Figs. S9–S12.

## 3.1 North America, Europe, and East Asia

Maps of the simulated atmospheric ammonia, ammonium and wet deposition of ammonium for North America, Europe and East Asia are shown in Figs. 3 and 4. The regions differ in both climatic conditions and their emissions of ammonia and other pollutants, which results in differences in the model performance across the different regions and species. The spatial patterns of the $NH_4^+$ wet deposition are well reproduced for the North American and and East Asian networks (R $\sim$ 0.8–0.9; Figs. 5 and 7). The model captures the peak wet deposition occurring in the US Midwest (Fig. 3c) and in Central and southwestern China (Fig. 4c), and reproduces the transition towards lower deposition in less agriculturally intensive areas. The measured wet deposition over the EMEP network is not captured as accurately ($R = 0.43$ in the FAN simulation; Fig. 6). The lower correlation is largely caused by 2–3 outlying stations with very high observed wet depositions (Fig. S9). The FAN and EDGAR simulations are also generally biased high ($\sim 25 - 40$ % of the observed mean wet deposition) especially in Western Europe. While the HTAP and CEDS simulations do not overestimate the average wet deposition in the EMEP dataset, which is consistent with their lower emitted totals, their spatial correlation coefficients remain low (0.47–0.51). The HTAP and CEDS simulations have the smallest average deposition bias also over Asia and North America, which might reflect the benefit of including regional emission data into the CEDS and HTAP inventories.

Similar to the global means, within each region $NH_3$ is biased low and $NH4^+$ is biased high in almost all simulations. For Europe, the bias of $NH_3$ ranges from a $\sim$40 % underestimation of the measurements (CEDS and HTAP) to a $\sim$10 % overestimation (EDGAR), whereas the $NH_4^+$ is overestimated by between 40 (CEDS, FAN, HTAP) and 70 % (EDGAR). The biases are even larger but more uniform for the U.S. (60–70 % underestimation of $NH_3$, 60–80 % overestimation of $NH_4^+$) and the performance is particularly poor for the East Asia, where all inventories predict unrealistically low $NH_3$ concentrations of below 0.5 $\mu$g N m$^{-3}$ over large areas; at the observed sites the negative bias is 80–90 % of the mean.

The CAM-chem version used in this study has previously been found to overestimate sulfate aerosol concentrations (Lamarque et al., 2012; Tilmes et al., 2015), possibly due to assumptions regarding the vertical distribution of sulfur dioxide emissions. This would lead into a positive bias in $NH_4^+$, and a correspondingly negative bias in $NH_3$, due to an overestimation of ammonium sulfates. The modeled wet deposition is likely to be more robust towards errors in the $NH_3/NH_4^+$ partitioning than the individual atmospheric measurements of these constituents, since both the gas-phase $NH_3$ and aerosol-phase $NH_4^+$ are scavenged by precipitation. This is consistent with the smaller biases and generally better performance of the simulated wet depositions compared to atmospheric $NH_3$ and $NH_4^+$ concentrations.

The differences between the simulations become more pronounced when evaluated in the temporal domain, as indicated by the Taylor plots (Figs. 5–7). A comparison of the monthly wet deposition and $NH_3$ emissions (Fig. 8) shows that in each simulation the temporal peaks in wet deposition coincide with the peaks in emissions in each of the networks, which suggests that the temporal differences in the different simulations can be traced to the temporal differences in $NH_3$ emissions. In FANv2, the seasonal variation in $NH_3$ emissions can be ascribed to both the $NH_3$ volatilization rate, which is driven by

**Figure 3.** Ammonia (panels a and d), ammonium (b, e), and wet deposition of ammonium (c, f) in the FAN simulation for North America and Europe (upper and lower panels, respectively). Observed values are shown by markers. Grey markers in panel f denote wet deposition observations from bulk samplers which are not used for calculating statistical parameters. The other simulations are shown in Figs. S6 and S7.





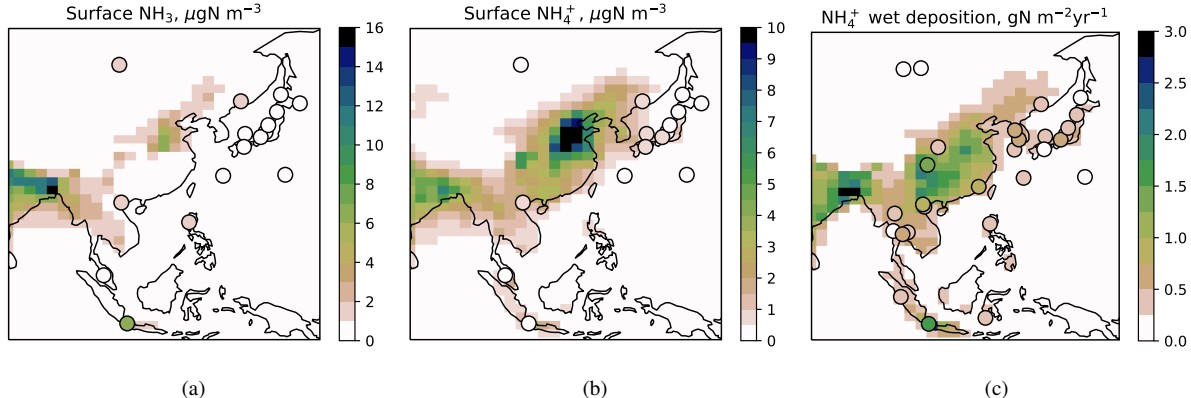

**Figure 4.** Ammonia (a), ammonium (b), and wet deposition of ammonium (c) in the FAN simulation for East Asia (shading) and in the EANET observations (markers). The other simulations are shown in Fig. S8.

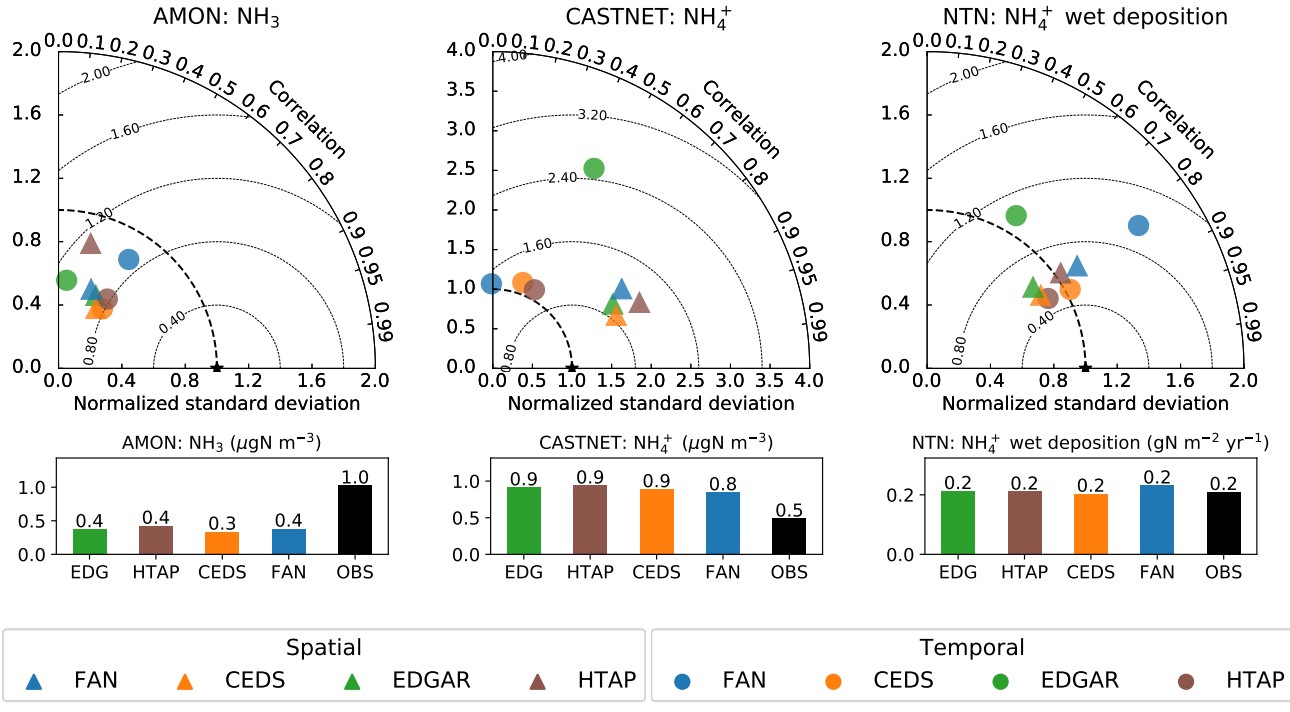

**Figure 5.** Taylor plots for the simulated surface $NH_3$ (left) and $NH_4^+$ (middle) concentrations and the $NH_4^+$ wet deposition (right) evaluated for the US observation networks (AMoN, CASTNET and NTN). The simulated and observed network-wide averages are shown in bar charts. The Taylor plots include both a temporal (first averaged in space) and spatial (first averaged in time) evaluation denoted by triangular and circular markers, respectively.





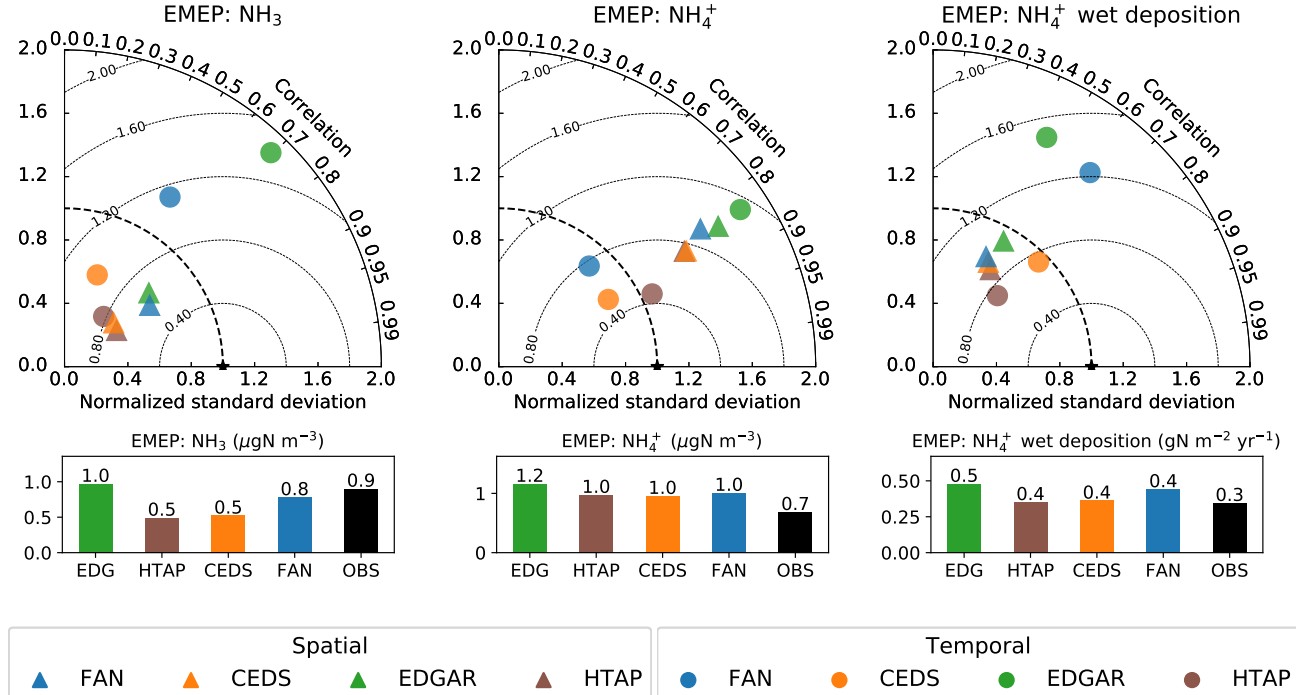

**Figure 6.** Same as Fig. 5, but for the EMEP network.

the simulated meteorology, and the fertilization timing, which is parameterized within the CLM crop model. The emission inventories prescribe these temporal variations on a monthly timescale.

The simulations broadly agree in the overall seasonal variation of the emissions, which in temperate climates results, in part, from the temperature contrast between the summer and winter months (Fig. 8). However, the simulations generally differ with

5 regard to the timing and magnitude of the springtime emission peak, a peak which can be attributed to springtime fertilization and possibly manure spreading. CEDS, EDGAR and FAN overestimate the springtime $NH_4^+$ wet deposition peaks to varying degrees; in FAN and especially EDGAR, the peak also occurs too early. The overall seasonality of the $NH_4^+$ wet deposition is best captured by the CEDS and HTAP simulations (Figs. 5–7), although compared to CEDS, HTAP has a tendency to underestimate the variability.

10 In summary, the four simulations show only small differences in their time-averaged spatial patterns, and their performance differ only slightly between the regions when evaluated against time-averaged observations (Figs. 5–7). In contrast, the simulations show distinct differences when compared temporally. This highlights temporal differences in the emissions; differences likely to originate in contrasting assumptions regarding the seasonality of agricultural activities. The temporal features of the observations are generally best reproduced using the CEDS and HTAP inventories, which incorporate regional datasets. In the





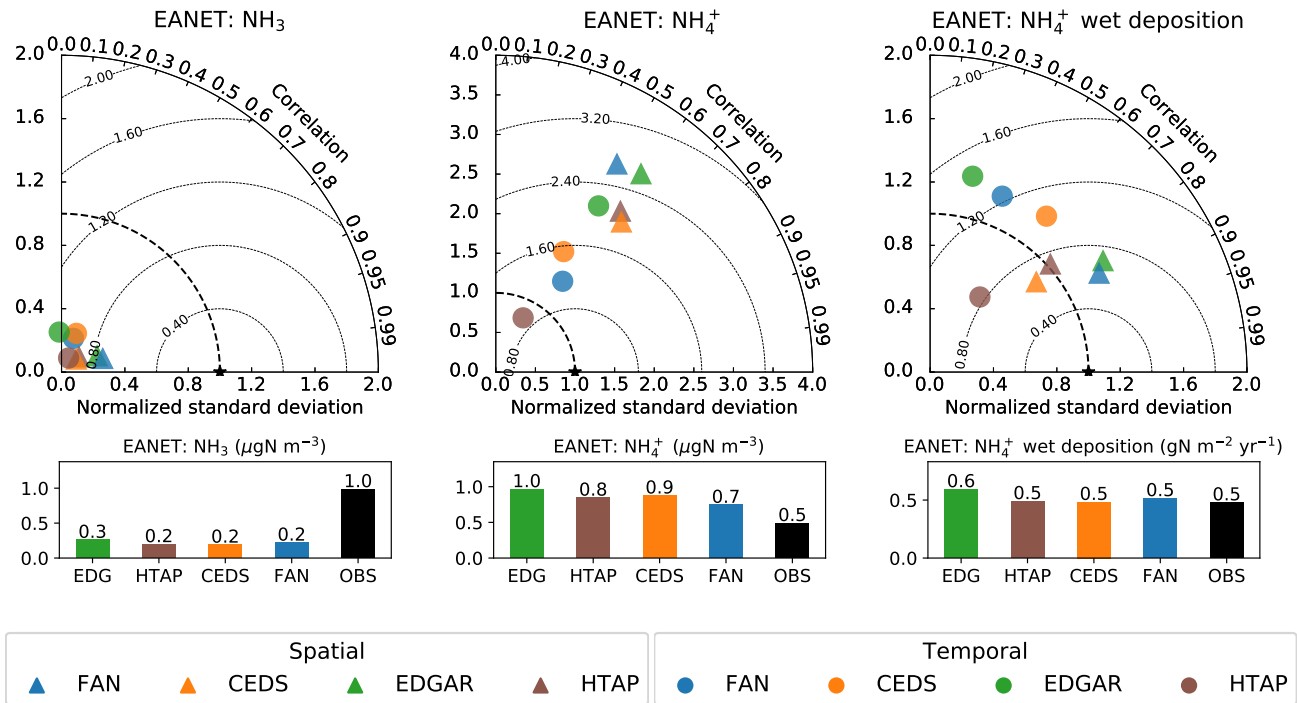

**Figure 7.** Same as Fig. 5, but for the EANET network.

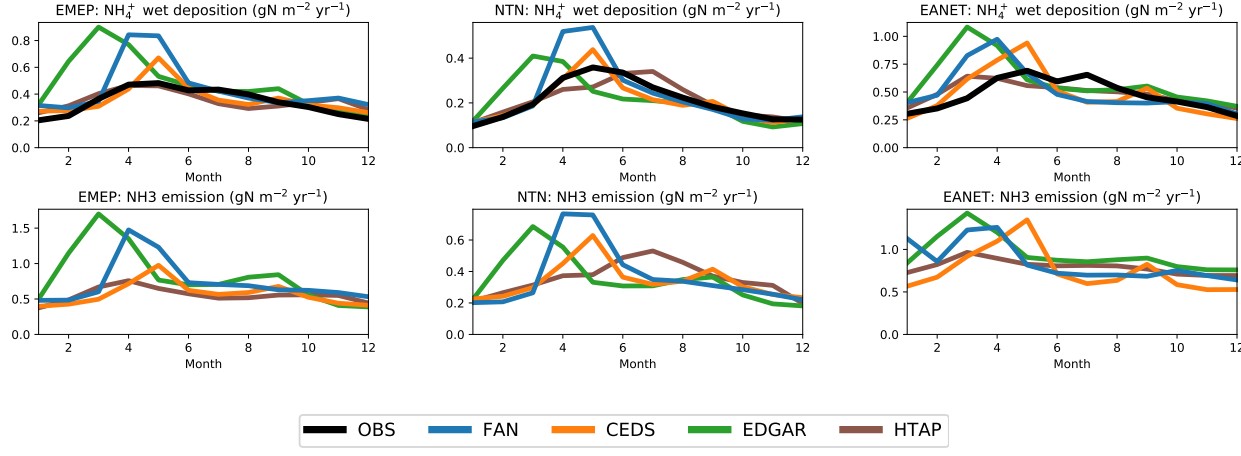

**Figure 8.** Seasonal profiles of simulated $NH_4^+$ wet deposition (upper panels) and and $NH_3$ emissions for the EANET, EMEP and NTN networks. All values are averaged over the years 2010–2015; the emission fluxes are evaluated for the observed sites.



FAN simulation the temporal correlation with the measurements generally lies between the EDGAR and the CEDS and HTAP simulations.

The results further suggest that both FAN and the inventories could be improved by assuming the fertilizer application is distributed more evenly over the growing season. All inventories, as well as FAN, would also benefit from a more realistic

representation of agricultural practices in the East-Asian region covered by EANET, where the observed wet deposition would be better captured by a more symmetric monthly variation in emissions peaking between May and July. This seasonal pattern is consistent with recent $NH_3$-focused global and regional studies (Xu et al., 2019; Zhang et al., 2018).

Regional fertilization practices could, in principle, be implemented in the CLM and thus within FANv2. This more detailed regional information would impact the agricultural nitrogen cycle simulated by the CLM, with consequences to the other

simulated aspects of the crop model including crop growth, harvest and N losses such as denitrification. Nevertheless, a central focus of the CLM as a component of an Earth system model is to simulate how ecosystem processes respond to the climate forcing, and it is therefore important to allow all simulated agricultural practices to change in response to the simulated climate. The direct coupling between FANv2 and the CLM means that FANv2 can benefit from improvements such as more detailed fertilization algorithms, should they be implemented in future versions of the CLM.

### 3.1.1   Africa

For Africa, the FAN-simulated $NH_3$ emissions (7.2 Tg N) differ markedly from the other inventories (2.1–2.4 Tg N; Table 1). While widespread observational data covering Africa are not available, the INDAAF dataset provides an opportunity to evaluate the predicted $NH_3$ emissions at stations located in Western and Central Africa. As noted in Section 2.5, due to limited data availability, the simulated results for 2010–2015 are here compared with measurements covering one or more years since

the year 2000 or later.

The comparison with the INDAAF data (Fig. 9) shows that both FAN and EDGAR emissions generally underestimate the average $NH_3$ concentration of 3.2 $\mu$g N m$^{-3}$ at the INDAAF sites. The EDGAR simulation predicts $NH_3$ concentrations mostly below 1.5 $\mu$g N m$^{-3}$ with a mean of 0.4 $\mu$g N m$^{-3}$, while the FAN simulation also underpredicts the $NH_3$ concentrations (simulated mean 1.0 $\mu$g N m$^{-3}$) but shows less bias in comparison to the available observations. As the CEDS and HTAP

emissions over Africa are based on the EDGAR inventory, the simulations using these inventories do not differ significantly from the EDGAR simulation.

The underestimation of $NH_3$ over Africa could be caused by inaccurate partitioning between the airborne $NH_3$ and $NH_4^+$ as noted in the previous section. While simulated $NH_4^+$ concentration in aerosol is indeed overestimated at one of the two INDAAF sites with sufficient data coverage for $NH_4^+$, the difference is far too small to explain the negative bias of gaseous

$NH_3$. The concentration of $NH_4^+$ in aerosol phase is generally low compared to $NH_3$, and the higher ammonia emissions in FAN compared with EDGAR mainly increase the concentration in the gas phase. Some of the increased $NH_3$ emissions in the FAN simulation are scavenged as $NH_4^+$ in the precipitation. As a result the $NH_4^+$ wet deposition is substantially higher in the FAN simulation than in EDGAR.



**Figure 9.** Modeled (shading) and observed (markers) for ammonia (left) and ammonium (middle) concentrations and wet deposition fluxes (right) for Africa in the FAN (upper) and EDGAR (lower panel) simulations. Modeled and observed values averaged over the stations are shown in the lower-left corner of each plot.



The EDGAR simulation underestimates the average wet deposition of $NH_4^+$ by about 50 %. The FAN simulation overestimates the wet deposition flux calculated from the observation by $\sim$ 10 %, but since the collection efficiency of the precipitation samplers is not perfect (see Section 2.5), the actual deposition flux is unlikely to be overestimated. The precipitation-weighted $NH_4^+$ concentrations in rainwater (Fig. S13) indeed show an average underestimation by about 25 %.

The modeled $NH_3$ concentrations at the INDAAF sites are consistent with the comparison of the column densities (Fig. 2), which show a dramatic difference between the FAN and EDGAR simulations over Africa north of the equator. The column density predicted by FAN is much closer to the IASI, although contrary to both FAN and the INDAAF observations, IASI shows a positive north to south gradient over Western Africa. The FAN simulation reproduces the observed pattern over Eastern Africa (Ethiopia and Kenya), while the values over Central Africa remain underestimated.

Biomass burning is generally an important source of $NH_3$ emissions over Africa (Bouwman et al., 1997; Whitburn et al., 2015). However, the relative contributions of biomass burning and agricultural $NH_3$ emissions differ between different ecosystems. The biomass burning emissions during the dry season are predominant in the forests and wet savannas, but volatilization of $NH_3$ from livestock wastes is the largest source in the Sahelian dry savannas (Adon et al., 2010), and it is indeed over the INDAAF sites in the Sahel region where FAN and the other simulations differ most markedly.

Fig. 10 compares the observed and simulated seasonal distributions of ammonia and its emission over the three dry savanna sites in the INDAAF dataset. Banizoumbou and Katibougou (Figs. 10b and 10c) show a two-peaked distribution with the highest $NH_3$ concentrations occurring during the transitions between the wet and dry seasons (boreal spring and autumn). A similar pattern has been observed for the concentration of nitrogen dioxide at the same sites (Ossohou et al., 2019). FAN captures the two-peaked seasonal pattern, although the concentration during the wet season is underestimated. The peaks are

offset by 1–2 months from that observed, because the simulated wet season starts earlier and ends later than observed. Different from the other two sites, the observed $NH_3$ concentration at the Agoufou site (Fig. 10a) remains high during the dry season, which is not reproduced.

The $NH_3$ emissions simulated by FAN at the Agoufou and Banizoumbou sites are mainly from manure handling and grazing livestock (not shown). No seasonal variation in the livestock N excretion is simulated over the region, and the seasonality of

the $NH_3$ emissions in Figs. 10a and 10b is therefore driven by changes in environmental conditions, especially soil moisture as suggested by earlier studies based on biogeochemical models (Delon et al., 2019) and empirical data (Hickman et al., 2018). The sharper peak in the $NH_3$ emission simulated for Katibougou in April (Fig. 10c) is due to fertilizer application on croplands, which coincides with the end of the dry season.

The seasonal patterns of wet deposition (Fig. 11) are strongly affected by the seasonality of precipitation. FAN predicts a

strong reduction in the deposition flux over the wettest months, similar to the seasonal variation of the gaseous $NH_3$. This, however, contrasts with the observed deposition fluxes which show only slight or no reduction for the month of peak precipitation.

The presence of significant livestock-originated $NH_3$ emissions over the Sahel region has been identified in earlier studies (Adon et al., 2010; Delon et al., 2010), and the non-pyrogenic origin of these emissions is consistent with the conclusion of

several studies based on satellite data (Whitburn et al., 2015; Van Damme et al., 2015; Someya et al., 2020). Similar to the





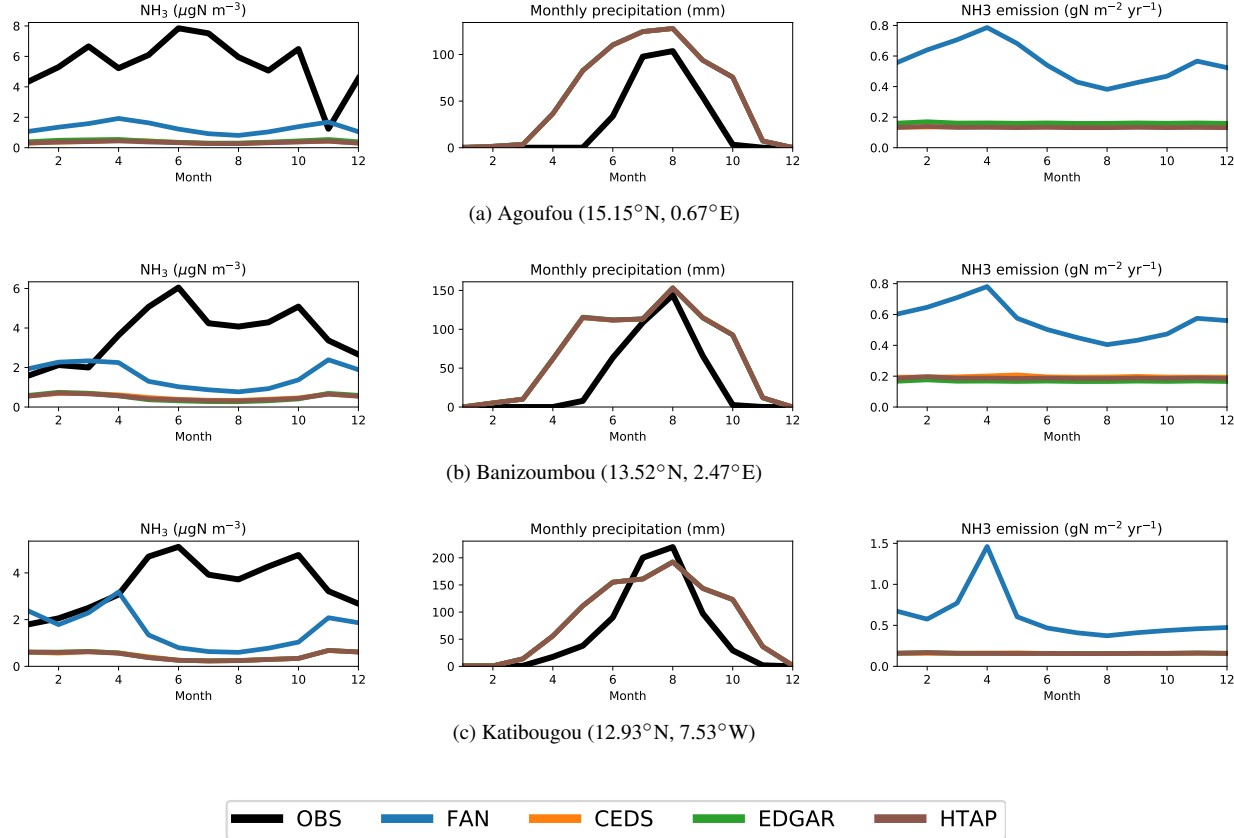

**Figure 10.** Simulated and observed ammonia concentration ($\mu$g N m$^{-3}$), monthly precipitation (mm) and the emission flux of ammonia (g N m$^{-3}$ yr$^{-1}$) on three dry savanna sites included in the INDAAF database.

assessment of Vet et al. (2014), we find that the current emission inventories underestimate the $NH_3$ and $NH_4^+$ concentrations and deposition over the Sahel region, and the comparison with IASI data furthermore suggests that a similar underestimation may exist also in other parts of Africa where livestock densities are high. The FAN simulation shows that some of the discrepancy can be reconciled using recent landuse and livestock datasets in combination with a process model which evaluates the 5  $NH_3$ volatilization as a function of environmental drivers.

### 3.1.2 Other regions

The FAN emissions are higher than in other inventories by ~20–35 % over India and by ~55-85 % over the Latin America. These regions are not covered by the monitoring networks included in this study and instead, we compare the simulations with annual and multiannual $NH_4^+$ wet deposition observations reported in literature for sites in India and Brazil (Table A1).

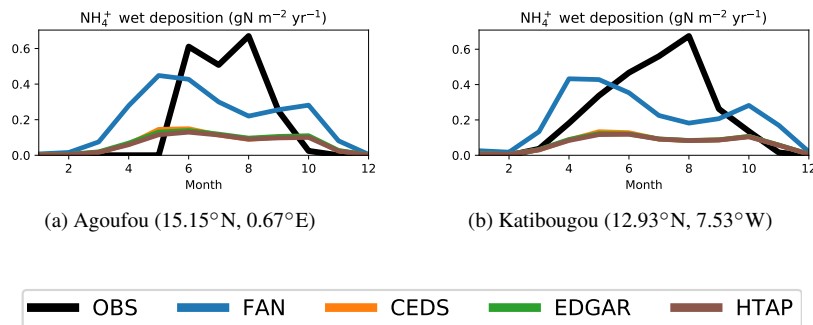

**Figure 11.** Simulated and observed wet deposition of ammonium ($\mu$g N m$^{-3}$ yr$^{-1}$) in Agoufou and Katibougou. The simulated wet deposition includes both scavenged aerosol-phase ammonium and the dissolved gaseous ammonia.

For the majority of the sites in India (Figs. 12a and S14), all simulations overestimate NH$_4^+$ wet deposition. The FAN simulation has the largest positive bias, as it has the highest NH$_3$ emissions of all the simulations (Table 1). The FAN simulation also has the highest spatial correlation with the measurements, although the overall agreement is modest in all simulations (R=0.14–0.30).

The NH$_4^+$ concentration in rainwater is overestimated in all simulations (Fig. S15), and thus, an overestimate in simulated precipitation is unlikely to explain the positive biases over India. Earlier studies (Dentener et al., 2006; Vet et al., 2014) have also found the NH$_4^+$ wet deposition to be overestimated in India, which could be caused by a systematic bias in emission inventories. The FAN simulation indeed overestimates the column-integrated NH$_3$ concentration with respect to the IASI data (Fig. 2) over parts of India, while EDGAR and the other simulations (Fig. S5) appear less biased despite the overestimated wet

depositions. As an alternative explanation, Singh and Kulshrestha (2012) suggest that the alkaline crustal aerosols typically present over the Indian subcontinent reduce the aerosol uptake and scavenging of NH$_3$ causing dry deposition to become the dominant removal pathway. This effect is not simulated by CAM-chem but could explain the apparent discrepancy between the reports of very high measured gaseous NH$_3$ concentrations, up to 70 $\mu$g m$^{-3}$ on a rural site (Singh and Kulshrestha, 2014), and the relatively low ($< 0.5$ gN m$^{-2}$yr$^{-1}$) wet deposition fluxes at several of the Indian sites.

Fig. 12b compares the simulated wet depositions to observations at four sites in Brazil. The deposition in the FAN simulation is on average slightly underestimated, but about 50 % higher than the other simulations (Figs. S16) and thus closer to the observations. The NH$_3$ column density in FAN also agrees better with the IASI data (Fig. 2), although the extent of the plume in northwestern Brazil is not captured. Similar to Africa, biomass burning is a significant source of NH$_3$ in South America, but more specific observations would be needed to differentiate between the agricultural and other emission sources. Thus, while

the observations are consistent with the higher NH$_3$ emissions predicted by FAN, the sparse geographical coverage and the

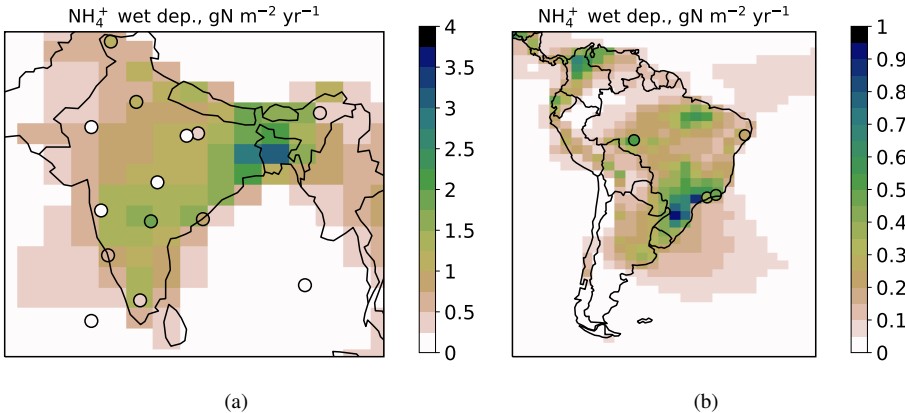

**Figure 12.** The modeled $NH_4^+$ wet deposition (gN m$^{-2}$yr$^{-1}$) over (a) India and (b) South America in the FAN simulation (shading) and in the observations listed in Table A1 (markers). The other simulations are shown in Figs. S14 and S16.

lack of co-located $NH_3$ and $NH_4^+$ observations make it difficult to draw definitive conclusions about the $NH_3$ emissions on a continental level.

Together, the $NH_3$ emissions in Africa, India and the Latin America comprise nearly half of the global total as simulated by FAN. Due to the scarcity of in-situ data, the emissions in these regions remain poorly constrained. Satellite retrievals of $NH_3$
offer an alternative data source for emission assessment (e.g. Chen et al., 2021); however, since only the gaseous ammonia is observable from satellites, this approach is sensitive to assumptions or model errors regarding in the lifetime and gas-aerosol partitioning of atmospheric $NH_3$.

### 3.2   Ammonium nitrate

The atmospheric concentration of ammonium nitrate depends on availability of ammonia and nitric acid vapor produced by
oxidation of nitrogen dioxide (Ansari and Pandis, 1998). The highest nitrate concentrations occur over populated areas in Asia, Europe and North America where the $NH_3$ emissions from intensive agriculture coincide with nitrogen oxide emissions from industrial and traffic sources (Fig. 13).

The FAN simulation is in good agreement ($R = 0.82$, $<10$ % mean bias) with the observed global patterns of aerosol-phase nitrate. The other simulations (Fig. S4) also reproduce the observations well (R = 0.74–0.76) although with slightly higher
biases (15-35 % of mean). However, comparing the networks separately (Fig. 14) reveals regional biases: The FAN simulation overestimates the North American (CASTNET) observations on average by $\sim$70 % and the European observations on average by $\sim$35 %, and underestimates the East Asian (EANET) observations on average by $\sim$50%. The other simulations show smaller underestimations for EANET but larger overestimations for CASTNET.

Some of the biases might be caused by inaccurate simulation of the gas or aqueous phase chemistry of the sulfur and nitrogen
oxides impacting the precursor concentrations for ammonium nitrates. In addition, besides ammonium nitrate, the measured

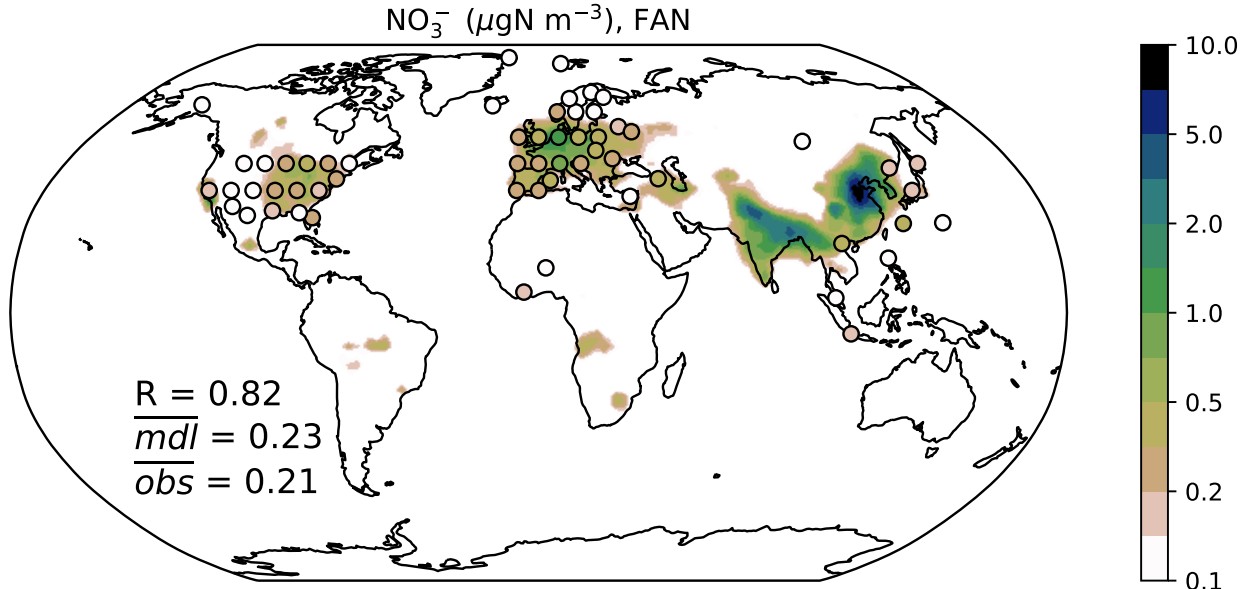

**Figure 13.** Average near-surface concentration of $NO_3^-$ ($\mu gN\ m^{-3}$) in the FAN simulation for 2010–2015. Observed values are indicated by markers. The density of observations has been reduced as in Fig. 1. The corresponding plots for the other simulations are shown in Suppl. Fig. S4.

nitrate aerosols may also form due to the interaction of nitric acid and sea salt or soil dust aerosols (Lee et al., 2008; Itahashi et al., 2016). The latter mechanisms are not simulated here. The omission of non-ammonium nitrate aerosols may explain the simulated negative bias seen in Fig. 13 at some coastal sites (e.g., several of the EANET stations) or continental sites (e.g., the INDAAF stations in the Sahel region). This, however, implies that the positive biases for the EMEP and CASTNET networks

would be even higher if CAM-chem included the nitrate in sea salt or dust particles.

As seen in Fig. 14, the temporal variation of $NO_3^-$ is in good qualitative agreement with the observations over North America and Europe in all simulations ($R = 0.8$–$0.9$), but its simulated amplitude (as measured by the standard deviation) is too high, especially for over the U.S. Over the East Asia, the temporal correlations show rather large variations between the simulations ($R = 0.45$–$0.8$) with FAN and CEDS performing best. It should be noted that the highest nitrate levels in the East Asia are

simulated to occur in China, which is not covered by the nitrate observations in EANET.

### 3.2.1 Effect of temporal resolution of emissions

For CASTNET, the FAN simulation has the lowest average nitrate concentration despite having the highest $NH_4^+$ wet deposition and second-highest total $NH_3$ emissions among the four simulations (Fig. 14 and Table 1). The FAN simulation also exaggerates the seasonal variation of $NO_3^-$ (Figs. 14 and S17) to a lesser extent than the other simulations. The seasonality

of the $NH_3$ emissions shown in Fig. 8 and the Taylor plot shown in Fig. 5 suggest this can not be attributed to FAN $NH_3$





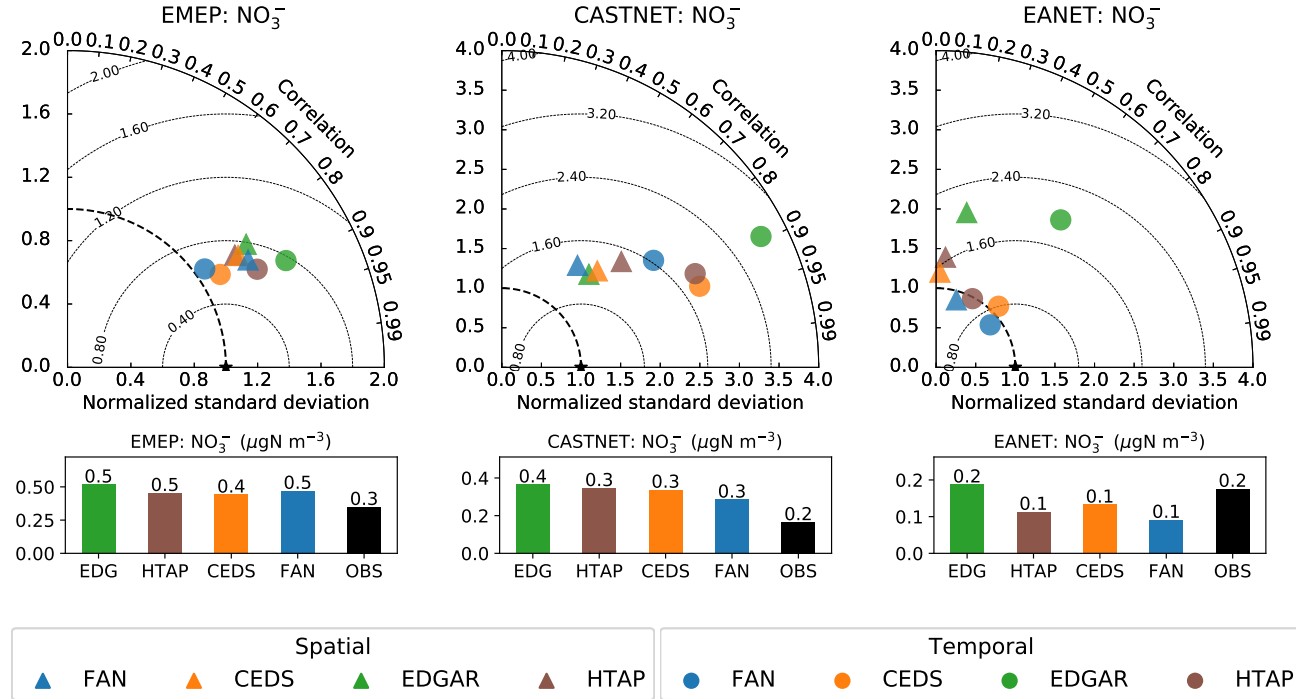

**Figure 14.** Taylor plots for the simulated $NO_3^-$ concentration in the EMEP, CASTNET and EANET networks. Average concentrations for 2010–2015 are shown in bar charts.

emissions having a significantly smaller seasonal amplitude than the other inventories. However, the differences between these simulations could be caused by the co-variation between $NH_3$ emissions and the chemical equilibrium which regulates the formation of ammonium nitrate aerosols: the cold, humid conditions which favor the aerosol-phase ammonium nitrate (Ansari and Pandis, 1998) are the least favorable for $NH_3$ volatilization (Vira et al., 2020). A set of one-year simulations was run to test

5   this hypothesis by averaging the FAN emissions to different time resolutions (hourly, daily, monthly, annually), but keeping the total emissions constant.

    Consistent with the above hypothesis, increasing the temporal resolution of emissions reduced the average $NO_3^-$ concentration throughout the world, although as seen in Fig. 15, the effect was geographically uneven. The largest absolute difference (up to ~0.5 $\mu$gN m$^{-3}$) between the simulations with hourly and monthly $NH_3$ emissions occurs in northern China. However,

10  if the difference is expressed as a fraction relative to the "monthly" run, the effect is largest in the Eastern and Central United States, where using hourly emissions decreases the mean $NO_3^-$ concentration by up to 25–30 % compared to monthly emissions. Nitrate aerosols are also noticeably reduced in parts of China, Eastern Europe and the Middle East. In Western Europe and India the temporal averaging of the emissions has only a small effect despite the relatively high $NO_3^-$ levels.





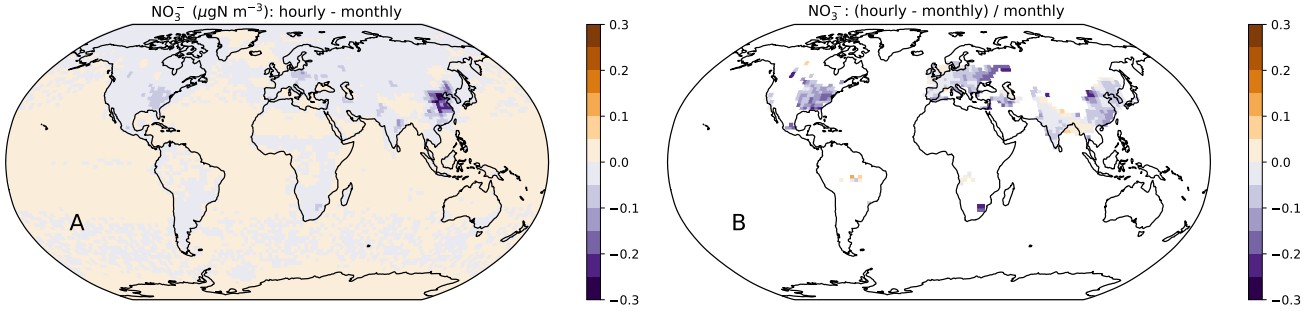

**Figure 15.** Difference in the 2010 mean aerosol $NO_3^-$ concentration between the simulations with monthly and hourly $NH_3$ emissions. Left: difference (hourly-monthly) in $\mu$gN m$^{-3}$; right: difference relative to the $NO_3^-$ concentration in the monthly simulation. The relative difference is evaluated for cells with annual mean concentration greater than 0.2 $\mu$gN m$^{-3}$.

**Table 3.** Bias (percent relative to observed) simulated $NO_3^-$ concentration for 2010 as affected by temporal averaging of $NH_3$ emissions. The biases are evaluated for the EANET, EMEP and CASTNET networks.

| | Bias (% observed mean) | | |
|---|---|---|---|
| Time averaging | EANET | EMEP | CASTNET |
| Yearly | +17 | +43 | +140 |
| Monthly | -30 | +31 | +80 |
| Daily | -34 | +28 | +70 |
| Hourly | -34 | +26 | +59 |
| | | | |
| Obs. mean, $\mu$gN m$^{-3}$ | 0.13 | 0.41 | 0.18 |

Comparison with observations (Table 3) confirms that increasing the time resolution of the emissions reduces the simulated nitrate concentrations. This reduces the positive simulation bias in the EMEP and CASTNET networks; over the EANET network the modeled bias changes from positive to negative. The largest reductions in nitrate occur from switching the emissions from the annual to the monthly resolution, and further, but more modest reductions occur by increasing the temporal resolution
5    to daily for all the measurement networks. Only over the CASTNET sites does a significant reduction occur when reducing the timescale of emissions from daily to hourly.

The effect of the emissions' temporal variation has previously not been tested systematically. Paulot et al. (2016) found that a prescribed, climatological diurnal variation of $NH_3$ emissions had only a small effect on surface-level nitrate. In contrast, Zhu et al. (2015) allowed the livestock $NH_3$ emissions in GEOS-Chem to vary proportionally to the temperature dependent effective
10   Henry constant for $NH_3$, and obtained reductions in the nitrate concentrations comparable to those in this study. It is possible that using the actual meteorology to drive the $NH_3$ emissions (as in Zhu et al. (2015) and this study) has a stronger effect on the





$NO_3^-$ concentration than an imposed diurnal cycle, since the co-variation of the $NH_3$ emission and nitrate formation is likely to be better resolved.

The positive model bias for nitrate over the Central and Eastern U.S. that occurs in CAM-chem (Lamarque et al., 2012) and other models has been connected to deficiencies in gas-phase chemistry (Heald et al., 2012) and aerosol scavenging (Luo et al.,

2019). Our results do not conflict with these earlier findings, since the positive bias is not fully resolved even using hourly $NH_3$ emissions. Nevertheless, the results indicate that some of the observed $NO_3^-$ bias could be caused by unresolved temporal variations of the $NH_3$ emissions due to a mechanism that stems from well-known thermodynamical properties of ammonia and ammonium nitrate.

## 4  Summary and conclusions

We have presented the first evaluation of a global chemistry-climate model simulation where ammonia emissions from both synthetic fertilizer and livestock are simulated interactively by a process model (FANv2). We compare the FANv2-enabled simulation with three conventional setups of the chemistry climate model CAM-chem, where the agricultural $NH_3$ emissions are prescribed based on the CEDS, EDGAR and HTAP emission inventories.

The simulations are evaluated against multi-component (atmospheric $NH_3$ and $NH_4^+$, and $NH_4^+$ wet deposition) in-situ

observations from European, East Asian and North American monitoring networks. When averaged over the 6-year period, the differences between these various simulations were relatively small, indicating that FANv2 offers a feasible alternative to the commonly used global emission inventories. The global patterns of wet deposition were especially well reproduced, whereas the gaseous ammonia and particulate $NH_4^+$ showed biases likely related to problems in simulating the chemistry of secondary inorganic aerosols in the bulk aerosol scheme in CAM-chem. Comparing the simulations temporally with a monthly

resolution revealed larger differences between the simulations using various emission inventories and suggested that a better characterization of fertilization practices would benefit both FANv2 and the emission inventories. The seasonal profiles used in CEDS and HTAP emissions captured the observed temporal patterns slightly better than FAN or the EDGAR inventory.

FAN and the emission inventories differ most over areas with scarce observational coverage. Over Africa, FAN predicts roughly 200–300 % higher $NH_3$ emissions than the EDGAR inventory or the HTAP and CEDS inventories derived from

EDGAR. Observations at the 4–7 sites (depending on species) included in the INDAAF network were consistent with the higher emissions in FAN, and the FAN simulation also agreed better with the $NH_3$ column densities retrieved from the IASI instrument over Africa. Regional ammonia emission inventories for Africa are currently not available, but our evaluation suggests that FANv2 may there capture the livestock-originated ammonia emissions better than the global inventories. Socioeconomic Pathways (SSPs) used in the Land Use Harmonization2 project (LUH2; Hurtt et al., 2020) all predict significant

livestock and population increases in Africa by 2100, indicating the importance of obtaining better observational constraints of ammonia emissions over Africa.





The $NH_4^+$ wet deposition patterns simulated using the FANv2 emissions were closer to to those measured at three sites in Brazil than the other assessed inventories, but measurements suggest FAN overestimates the emissions in India. However, the lack of co-located observations of atmospheric $NH_3$ or $NH_4^+$ make the the comparisons over India and Brazil less conclusive.

Finally, we evaluate the simulations against observations of aerosol-phase nitrate and show that the simulated ammonium nitrate concentrations are, even on a yearly level, sensitive to meteorology-driven daily and hourly variations in the $NH_3$ emissions. While the effect is geographically variable, our results suggest that some of the overestimation of nitrate aerosols over the eastern United States in the CAM and other models may be explained by unresolved temporal variations in the emissions of $NH_3$.

In conclusion, in simulating ammonia and ammonium concentrations over regions with detailed regional emission inventories, the inventories based on these details (HTAP, CEDS) capture the atmospheric concentrations and their seasonal variability the best. However, they can not simulate the daily to interannual variations in emissions due to meteorological variability. This variability may be substantial (e.g., Sutton et al., 2013) and also important for simulating nitrate aerosols. As a process model, FANv2 is capable of simulating this variability, although we do not specifically examine it in this paper. In a larger context, agriculture plays an important role in the global cycles of carbon and nitrogen, and to capture its impact it is essential to simulate agriculture dynamically within an Earth system model.

In the future, we intend to integrate the FANv2 emission model more tightly to the nitrogen cycling simulated by the Community Land Model. This would allow FANv2 to simulate how the volatilization losses affect the biogeochemistry of agricultural ecosystems. Conversely, it would allow FANv2 to take advantage of possible advances in representation of agricultural practices such as tillage and irrigation, and to estimate their possible positive or negative impacts on the ammonia emissions.



*Code and data availability.* The Community Earth System Model, including the Community Land Model (CLM) is available at www.cesm.ucar.edu. The modified version of CLM used in this paper is available at https://doi.org/10.5281/zenodo.3841776. The full modified version of CESM, including changes to CAM and the coupler interface, requires access to the CAM development repository which can be granted by UCAR upon agreement with the terms of use. The $NH_3$ emissions simulated by FAN and the FAN-specific input data are available at

at https://doi.org/10.5281/zenodo.3841723; other model outputs are available upon a reasonable request from the authors. The other model input and all observational data were obtained from public databases and datasets. The $NH_3$ emission inventories used in this study are available at http://data.europa.eu/89h/jrc-edgar-v432-ap-gridmaps (EDGAR v4.3.2), https://edgar.jrc.ec.europa.eu/dataset_htap_v2 (HTAP v2.2), and in Hoesly et al. (2017) for the CEDS. The CASTNET observations are available at https://www.epa.gov/castnet. The AMoN and NTN observations are available at http://nadp.slh.wisc.edu/data/. The EMEP observations are available at http://ebas.nilu.no/, the EANET obser-

vations are available at https://www.eanet.asia/, and the INDAAF observations are available at https://indaaf.obs-mip.fr/ upon registration. The IASI data are published in Van Damme et al. (2018b). All websites were accessed on 22 June 2021.

*Author contributions.* JV and PH designed the experiments; JV performed the simulations and analyzed the results with contributions from PH, MO and CGL. JV and PH wrote the manuscript with contributions from MO and CGL.

*Competing interests.* The authors declare no competing interests.

*Acknowledgements.* This work was funded in part by the Department of Energy (#DE-SC0016361) and in part supported by the National Center for Atmospheric Research, which is a major facility sponsored by the National Science Foundation under Cooperative Agreement No. 1852977. Computing resources (doi:10.5065/D6RX99HX) were provided by the Climate Simulation Laboratory at NCAR's Computational and Information Systems Laboratory, sponsored by the National Science Foundation and other agencies. We thank the Joint Research Centre for the EDGAR emission dataset made available at https://data.europa.eu/doi/10.2904/JRC_DATASET_EDGAR. We thank the National

Atmospheric Deposition Program for the access to the NTN and AMoN data sets, and the U.S. Environmental Protection Agency Clean Air Markets Division for the CASTNET data set. We also thank Network Center for EANET for the EANET Data on the Acid Deposition in the East Asian Region, and the Cooperative Programme for Monitoring and Evaluation of the Long-range Transmission of Air Pollutants in Europe (EMEP) for provision of the European air pollution monitoring data. The authors thank Will Wieder and Erik Kluzek for assistance with the CESM model and Marje Prank for comments on the manuscript. The Taylor diagrams in this paper were produced using the Python

module taylorDiagram.py by Yannick Copin.



# Appendix A: $NH_4^+$ wet deposition measurements from Brazil and India

**Table A1.** $NH_4^+$ wet deposition measurements from Brazil and India.

| Country | Site | Longitude | Latitude | Years | Type | $NH_4^+$ deposition, $g\,N\,m^{-2}\,yr^{-1}$ | Reference |
|---|---|---|---|---|---|---|---|
| Brazil | Mundau | -36.37 | -8.87 | 2012–2013 | rural | 0.24 | Deusdará et al. (2017) |
| | Rondonia | -61.93 | -10.01 | 2002[1] | rural | 0.52 | Trebs et al. (2006) |
| | Cunha | -45.07 | -23.25 | 2001–2002 | rural | 0.38 | Vet et al. (2014) |
| | Rio de Janeiro State | -43.03 | -22.66 | 2008–2009 | multiple[2] | 0.48 | de Souza et al. (2015) |
| | | | | | | | |
| India | Hudegadde | 74.54 | 14.36 | 2006–2008 | rural | 0.63 | Kulshrestha et al. (2014) |
| | Hyderabad | 78.50 | 17.50 | 2005–2008 | urban | 1.78 | Kulshrestha et al. (2014) |
| | Delhi | 77.15 | 28.53 | 2013 | urban | 1.05 | Singh et al. (2017) |
| | Jaunpur | 82.85 | 25.62 | 2013 | rural | 0.32 | Singh et al. (2017) |
| | Minicoy | 73.00 | 8.30 | 2000–2007 | rural | 0.07 | Rao et al. (2014) |
| | Mohanbari | 95.00 | 27.48 | 2000–2007 | rural | 0.40 | Rao et al. (2014) |
| | Portblair | 92.72 | 11.67 | 2000–2007 | rural | 0.15 | Rao et al. (2014) |
| | Srinagar | 75.15 | 33.63 | 2000–2007 | urban | 1.11 | Rao et al. (2014) |
| | Allahabad | 81.10 | 27.20 | 2000–2007 | urban | 0.13 | Rao et al. (2014) |
| | Jodhpur | 80.20 | 27.10 | 2000–2007 | urban | 0.04 | Rao et al. (2014) |
| | Nagpur | 80.60 | 25.60 | 2000–2007 | urban | 0.14 | Rao et al. (2014) |
| | Pune | 80.00 | 24.90 | 2000–2007 | urban | 0.13 | Rao et al. (2014) |
| | Visakahpatnam | 80.40 | 24.00 | 2000–2007 | urban | 1.00 | Rao et al. (2014) |
| | Kodaikanal | 78.30 | 14.90 | 2000–2007 | rural | 0.29 | Rao et al. (2014) |

[1]Extrapolated from a campaign using climatological precipitation data; [2]Average of two montane and one peri-urban site.



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
