# Peer review of "Evaluation of interactive and prescribed agricultural ammonia emissions for simulating atmospheric composition in CAM-Chem"

_Atmospheric Chemistry and Physics, 2021_

## Author Response (AR1)

We thank the anonymous referees 1 and 2 for their constructive comments on our manuscript. Our responses to the reviewer comments are given below.

We have prepared a revised manuscript which addresses the referee comments; in addition, we have corrected inconsistencies in the numbering of the subsections of Section 3. Furthermore, the coordinates of one of the INDAAF measurement sites (Agoufou) were displaced by about 250 km, and we have redrawn the figures to represent the correct location. The practical difference is very small.

Referee comments are shown in italic.

Responses to Referee #1

Page 10 Line 2. Referring to Figures S9-12 here is a bit confusing as these belong to section 3.1 below. I recomend refering those figures within section 3.1.

Thank you for the suggestion. We have changed the text to reference the figures from within Section 3.1.

Page 18. Figure 10. Monthly precipitation from observations is compared to FAN or HTAP?

The precipitation is practically identical in all simulations; a note of this has been added to the caption.

Page 20 Line 14. FigS4 should be FigS17?

We were actually referring to Fig. S4, which shows a similar map as Fig. 13 but for the CEDS, EDGAR and HTAP runs. We realize that the order of supplementary figures is confusing and have moved Fig. S4 to appear as Fig. S16.

Page 25 Line 1 "to to"

Page 24 Line 3 "the the"

Corrected.

Responses to Referee #2

Page 19, line 10-15: the impact of crustal material on the partitioning of NH3 to the aerosol phase and subsequently on dry deposition flux is expected to also affect NH3 levels over Africa and lead there to a higher underestimate. This has to be somehow discussed.

We agree. A brief discussion has been added in Section 3.2.

Page 21, lines 12-13: (despite) Here, it could be mentioned again the involvement of sulfate aerosol that is overestimated (more ammonium sulfate formed in the model) as stated earlier by the authors, so it seems that modelled aerosol is more acidic than observed and it make sense to have less nitrate partitioning in the aerosol phase (see for instance Nenes et al., Atmos. Chem. Phys., 21, 6023–6033, 2021). I would expect that the result will be different

when the model will account for crustal and sea salt nonvolatile cations, bringing nitrate simulations closer to observations? Changing temporal resolution of NH3 emissions is also affecting the acidity of the aerosol and thus the partitioning of nitrate to it.

We agree that our results could be affected by the model limitations as suggested by the reviewer. We note that the model overestimates the sulfate concentrations in CASTNET in the summer while it overestimates the nitrates in winter. There are a large number of possible reasons for these biases including biases in SO2 emissions, SO2 oxidation, in boundary layer meteorology or in aerosol partitioning itself. Nevertheless, it will be interesting to see whether the effect of interactive NH3 emissions will be similar if coupled to the recently introduced CAM5-Chem-MAM7-MOSAIC model version (Zaveri et al., 2021), which includes a detailed thermodynamic model for gas-aerosol partitioning as well as a revised treatment of sulfate aerosols as a part of the CAM5 physics package.

We have added simulated and observed sulfates to the monthly distributions shown in Fig. S17, and added the following text:

"The overestimation of sulfate aerosols (Section 3.1) could disturb the gas-aerosol partitioning of nitric acid, but this would be expected to rather result in a negative bias in the simulated nitrates (Feng and Penner, 2007; Nenes et al., 2021). The annually averaged effect in our simulations is likely to be small due to the opposite seasonal distribution of the model biases for nitrate and sulfate (Fig. S17)."

Page 17, line 21: do you have an explanation why NH3 concentrations behave differently at the Agourou site than the other sites?

The difference might be caused by the ambient conditions or local emissions, but both explanations are at this point speculative due to lack of detailed local information. We have added the following discussion:

"Different from the other two sites, the observed NH3 concentration at the Agoufou site (Fig. 10a) remains high during the dry season, which is not reproduced. The reasons for this difference are unclear. Agoufou records a lower yearly rainfall than Banizoumbou and Katibougou, but in addition, Delon et al. (2015) note that the surrounding area is heavily grazed during the dry season due to the proximity of a permanent pond. We therefore cannot exclude the possibility that the observed seasonal variation would reflect the local NH3 emissions."

Page 20, line 8: I propose changing title to 'nitrate aerosol' since measurements do not necessarily concern NH4NO3 as also stated by the authors at the bottom of this page.

Done.

Page 5, line 13: please specify if it is nitrate observations in bulk aerosol, PM10 or PM2.5

We have added the following sentence:

"The nitrate and ammonium concentrations are measured within bulk aerosol (without restrictions in particle size), except in the EMEP dataset, which depending on the station includes measurements in both PM2.5 and bulk aerosol."

Page 7, line 21: 'of by' I would keep 'of'

We agree.

Page 10, lines 28-29: here you could clarify that wet deposition flux of NH3 and NH4+ are measured together as NH4+

Done.

Page 17, line 8: specify north of the equator

The sentence has been modified to read "...Africa between the Equator and approximately 15° N.".

Page 17, line 9: Central and West Africa

Page 20, line 6: I would remove 'regarding'

Page 25, line 1: 'to to' remove one

Page 25, line 3: ' the the' remove one

Done.

**References**

- Delon, C., Mougin, E., Serça, D., Grippa, M., Hiernaux, P., Diawara, M., Galy-Lacaux, C., Kergoat, L., 2015. Modelling the effect of soil moisture and organic matter degradation on biogenic NO emissions from soils in Sahel rangeland (Mali). Biogeosciences 12, 3253–3272. https://doi.org/10.5194/bg-12-3253-2015
- Zaveri, R.A., Easter, R.C., Singh, B., Wang, H., Lu, Z., Tilmes, S., Emmons, L.K., Vitt, F., Zhang, R., Liu, X., Ghan, S.J., Rasch, P.J., 2021. Development and Evaluation of Chemistry-Aerosol-Climate Model CAM5-Chem-MAM7-MOSAIC: Global Atmospheric Distribution and Radiative Effects of Nitrate Aerosol. J. Adv. Model. Earth Syst. 13, e2020MS002346. https://doi.org/10.1029/2020MS002346